# Relating Hopfield Networks to Episodic Control

**Hugo Chateau-Laurent** ⓘ*
Inria centre of the University of Bordeaux, France
IMN, CNRS UMR 5293, France
LaBRI, CNRS UMR 5800, France
`hugo.chateaulaurent@gmail.com`

**Frédéric Alexandre** ⓘ
Inria centre of the University of Bordeaux, France
IMN, CNRS UMR 5293, France
LaBRI, CNRS UMR 5800, France
`frederic.alexandre@inria.fr`

## Abstract

Neural Episodic Control is a powerful reinforcement learning framework that employs a differentiable dictionary to store non-parametric memories. It was inspired by episodic memory on the functional level, but lacks a direct theoretical connection to the associative memory models generally used to implement such a memory. We first show that the dictionary is an instance of the recently proposed Universal Hopfield Network framework. We then introduce a continuous approximation of the dictionary readout operation in order to derive two energy functions that are Lyapunov functions of the dynamics. Finally, we empirically show that the dictionary outperforms the Max separation function, which had previously been argued to be optimal, and that performance can further be improved by replacing the Euclidean distance kernel by a Manhattan distance kernel. These results are enabled by the generalization capabilities of the dictionary, so a novel criterion is introduced to disentangle memorization from generalization when evaluating associative memory models.

## 1   Introduction

Episodic memory is the ability to remember information about a specific situation. An influential model of episodic memory is the Hopfield Network (1), a recurrent associative memory that can learn a pattern in one shot and recall it, given some partial or noisy cue. Some important limitations have been addressed with the development of differentiable continuous Hopfield Networks (2) and their connection to deep learning (3; 4), thus providing a renewed interest to the field of associative memory. Episodic memory has also been studied as an efficient way to control reinforcement learning in so-called episodic control, particularly in the initial steps of learning (5), but no explicit link has been made between Hopfield Networks and control algorithms. Such a link could lead to the development of more efficient controllers and memory models. It could also shed light on how the hippocampus, the seat of episodic memory in the brain, contributes to behavior.

In this paper, a novel connection is established between the fields of associative memory and reinforcement learning. It is shown in Section 2 that the differentiable neural dictionary (DND) introduced in the context of episodic control (6) as a rapid way to store and retrieve experiences, is mathematically close to the Hopfield Network. Retrieval from a DND can indeed be decomposed into

---

*Currently a postdoc at CerCo, CNRS UMR 5549, Université de Toulouse, France.

38th Conference on Neural Information Processing Systems (NeurIPS 2024).

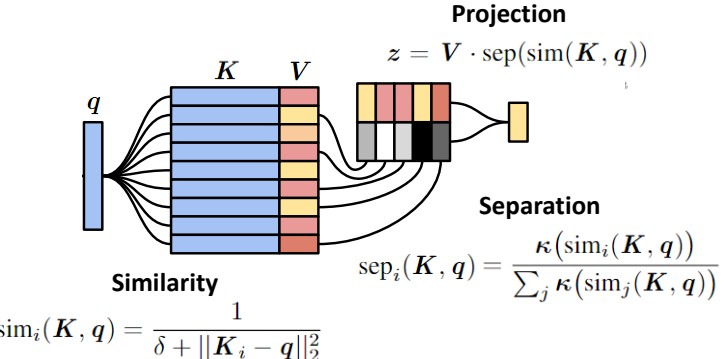

**Projection**
$$\boldsymbol{z} = \boldsymbol{V} \cdot \text{sep}(\text{sim}(\boldsymbol{K}, \boldsymbol{q}))$$

**Separation**
$$\text{sep}_i(\boldsymbol{K}, \boldsymbol{q}) = \frac{\kappa\big(\text{sim}_i(\boldsymbol{K}, \boldsymbol{q})\big)}{\sum_j \kappa\big(\text{sim}_j(\boldsymbol{K}, \boldsymbol{q})\big)}$$

**Similarity**
$$\text{sim}_i(\boldsymbol{K}, \boldsymbol{q}) = \frac{1}{\delta + ||\boldsymbol{K}_i - \boldsymbol{q}||_2^2}$$

Figure 1: Differentiable Neural Dictionary lookup as retrieval from a Universal Hopfield Network.

similarity, separation and projection operations, just as any instance of the general Universal Hopfield Network framework (UHN) (7) recently proposed to encompass both classical and recent models of associative memory. In a DND, similarity scores are computed using the Euclidean distance and separated with a $k$-nearest neighbor algorithm. The projection operation of DND is the same as for all existing UHN instances, including the traditional Hopfield Network. DND can thus be thought of as a single-shot associative memory model, just like Hopfield Networks and their modern continuous variants. These models can often be defined by an energy function which decreases as memories are retrieved. We show that energy functions can also be derived for the recall operation of the DND.

In Section 3, experiments are conducted to compare the DND to state of the art associative memory models in memorization tasks. It is shown that the DND outperforms concurrent models in generalization tasks. Its performance is further improved by replacing the Euclidean distance by the Manhattan distance, as predicted by the original UHN study (7). A new criterion is introduced to assess performance, and it is found that the $k$-nearest neighbor separation of DND favors generalization over memorization, as compared to the simpler Max separation function.

## 2 Differentiable Neural Dictionary as a Hopfield Network

The UHN framework encompasses a family of associative memory models in which retrieval is performed by computing the similarity between a query **q** and keys **K** (sim function), separating the similarity scores with a function sep, then projecting the results to the output space with some value matrix **V**:

$$\boldsymbol{z} = \boldsymbol{V} \cdot \text{sep}(\text{sim}(\boldsymbol{K}, \boldsymbol{q})). \tag{1}$$

In the original binary Hopfield Network (1), the similarity function is the dot product and sep is the identity function. Modern continuous variants have been proposed (8) that improve storage capacity by using more elaborate separation functions such as Softmax (4) to push apart memory attractors.

On the other hand, Neural Episodic Control is a reinforcement learning architecture that introduces the DND as a way to store associations between sensory observations and Q-value estimates. The reading operation of the DND is:

$$z = \sum_{i=1}^{k} \phi_i \boldsymbol{w}_i, \tag{2}$$

where $\boldsymbol{w}$ contains the normalized inverse distances between the query and the $k$ nearest observation keys, and $\phi$ contains the values of nearest observations (i.e. their Q-value estimates). The inverse distances are computed using the following kernel function:

$$\text{sim}_i(\boldsymbol{K}, \boldsymbol{q}) = \frac{1}{\delta + ||\boldsymbol{K}_i - \boldsymbol{q}||_2^2}, \tag{3}$$

with **K** the observation keys, **q** the query, and $\delta = 10^{-3}$. The $k$-nearest neighbor function can be written as:

$$\text{sep}(\boldsymbol{x}) = \boldsymbol{\kappa}(\boldsymbol{x}) / \sum_i \kappa_i(\boldsymbol{x}) \tag{4}$$

$$\boldsymbol{\kappa}_i(\boldsymbol{x}) = \begin{cases} \boldsymbol{x}_i & \text{if } \boldsymbol{x}_i \text{ is among the top } k \text{ values of } \boldsymbol{x}, \\ 0 & \text{otherwise.} \end{cases} \tag{5}$$

We can thus rewrite Equation 2 as:

$$z = \phi \cdot \text{sep}(\text{sim}(\boldsymbol{K}, \boldsymbol{q})), \tag{6}$$

which closely resembles the reading operation of UHN (Equation 1).

Note that the Euclidean similarity function of UHN (7) is the same as the kernel function of Neural Episodic Control (6):

$$\text{sim}_i(\boldsymbol{K}, \boldsymbol{q}) = \frac{1}{\delta + \sum_j (\boldsymbol{K}_{ij} - \boldsymbol{q}_j)^2} \tag{7}$$

$$= \frac{1}{\delta + \left(\sqrt{\sum_j (\boldsymbol{K}_{ij} - \boldsymbol{q}_j)^2}\right)^2} \tag{8}$$

$$= \frac{1}{\delta + ||\boldsymbol{K}_i - \boldsymbol{q}||_2^2} \tag{9}$$

In sum, the kernel function of DND acts as a similarity function and is equivalent to the Euclidean similarity function of UHN. Furthermore, the $k$-nearest neighbor algorithm sparsifies the result of the similarity function by cancelling the contribution of the most distant experiences. The output of the algorithm is then normalized before being projected to the value space. This constitutes a novel separation function for the UHN framework, we call it $k$-Max. It is worth noting that this separation function is similar to applying a threshold on the similarity function, like what is done in the sparse distributed memory model (9), which has also been cast as a UHN (7). The only difference is that the threshold for sparse distributed memory is fixed, while it must be dynamic for selecting a constant number $k$ of neighbors. Furthermore, with $k = 1$, the separation function is equivalent to the Max separation function of UHN. The remaining difference between DND and other UHN instances is that the output of the DND is a scalar value, while UHN models can store vector values. In a modification of DND (10), multidimensional values have been stored. In fact, the DND can simply be extended with a matrix of value vectors $\boldsymbol{V}$. Equation 6 thus becomes:

$$\boldsymbol{z} = \boldsymbol{V} \cdot \text{sep}(\text{sim}(\boldsymbol{K}, \boldsymbol{q})). \tag{10}$$

An abstract energy function has been derived for UHN models (7). It is defined as:

$$E(\boldsymbol{K}, \boldsymbol{v}) = \sum_i \frac{1}{2} \boldsymbol{v}_i^2 - \int \text{sep}\left[\sum_j \text{sim}(\boldsymbol{K}_{i,j}, \boldsymbol{v}_i)\right], \tag{11}$$

with $\boldsymbol{v}$ the activity of value neurons which are initialized with the query $\boldsymbol{q}$, and updated to produce the output pattern $\boldsymbol{z}$ (in the case of autoassociation where $\boldsymbol{V} = \boldsymbol{K}^\top$).

The energy function requires the gradient of the separation function to be nonzero, which is not the case for $\boldsymbol{\kappa}$ (as defined in Equation 5). A few adjustments thus need to be made to $\boldsymbol{\kappa}$ in order to derive the energy function for the DND. Let $\sigma(x)$ denote the Sigmoid function, defined as:

$$\sigma(x) = \frac{1}{1 + e^{-x}} \tag{12}$$

An adjusted Sigmoid function with threshold $\Theta$ and steepness parameter $\beta > 0$, is used to define a continuous approximation of $\boldsymbol{\kappa}$:

$$\boldsymbol{\kappa}_i(\boldsymbol{x}) = \boldsymbol{x}_i \sigma\left(\beta(\boldsymbol{x}_i - \Theta)\right) \tag{13}$$

$$\boldsymbol{x}_i = \sum_j \text{sim}(\boldsymbol{K}_{i,j}, \boldsymbol{v}_i). \tag{14}$$

A second Sigmoid function is used to count the number of selected dimensions ($\boldsymbol{\kappa}/\boldsymbol{x}$ entries higher than $1/2$) and adjust $\Theta$ to ensure there are only $k$ of them:

$$\Theta^{t+1} = \Theta^t + \alpha\left[-k + \sum_{i=1}^n \sigma\left(\beta_k\left(\frac{\boldsymbol{\kappa}_i}{\boldsymbol{x}_i} - \frac{1}{2}\right)\right)\right], \tag{15}$$

where $0 < \alpha < 1$ governs the threshold dynamics and $\beta_k$ controls the steepness of the second Sigmoid. In Appendix A, we show that $\Theta$ is convergent with $\alpha < \frac{16}{n\,\beta_k\,\beta}$. Equation 13 has a nonzero gradient, and as $\beta$ and $\beta_k$ grow to infinity, it approaches Equation 5. Hence, it can be used for the separation function of Equation 11, providing DND with an energy function. A second energy function is the update of $\Theta$:

$$E = \Delta(\Theta) = \alpha\left[-k + \sum_{i=1}^n \sigma\left[\beta_k\left(\sigma\left(\beta(\boldsymbol{x}_i - \Theta)\right) - \frac{1}{2}\right)\right]\right] \tag{16}$$

## 3 Associative memory performance of the Differentiable Neural Dictionary

In the previous section, the differentiable neural dictionary of Neural Episodic Control has been shown to be a Universal Hopfield Network. In principle, DND can thus be used as an associative memory. In this section, the MNIST, CIFAR10 and Tiny ImageNet datasets are used to test the robustness and capacity of DND as an associative memory model, using the same methods as for the other UHN instances (7) unless otherwise mentioned[2].

Example reconstructions of images by DND are shown in Figure 2, as well as Figures 8 and 7. Memories are separated by keeping the $k$-nearest neighbors only ($k$-Max separation function). For simplicity, the $kd$-tree of the original Neural Episodic Control implementation is not used, nor is the continuous $k$-Max version that makes use of Equation 13. Instead, all similarity scores are computed and those not selected are zeroed out (Equation 5). In Figure 2, using $k = 50$ like in the original Neural Episodic Control publication (6) gives an output from which the original image can be recognized, but the model is unable to properly separate memories and the resulting output is blurry. The Max separation function is equivalent to selecting the nearest neighbor ($k = 1$) and provides a much clearer output. In fact, Max is the best separation function benchmarked for UHN (7). Even when the output is blurry, the $k > 1$ models seem to properly capture the statistics of the dataset, such as the fact that central pixels are globally more active than those on the borders. Recall accuracy is typically assessed in absolute terms, by checking that the difference between output and response is below a threshold (7). It is worth exploring whether the statistical modeling capacities of $k > 1$ can consistently improve performance with this criterion, despite the Max function having theoretically unbounded capacity with respect to the dimensionality of the query (7). On the other hand, a new criterion will be introduced to evaluate recall in relation to other stored patterns, given that the main function of episodic memory is to reconstruct information corresponding to a particular situation (memorization) rather than to generalize.

Throughout the paper, we distinguish between two key aspects of associative memory models, adhering to the terms of (7): capacity and retrieval. Capacity refers to the number of unique images (or memories) that can be stored while maintaining accurate recall. Retrieval, on the other hand, focuses on the model's performance in recalling these stored images when they are presented with incomplete or noisy cues. It measures the resilience of memory recall in the presence of distortion.

### 3.1 Capacity with different functions

In DND, the similarity between memories and the query is computed using a Euclidean function (Equation 3). While rarely used in associative memory models, this function was found to outperform the more common dot product (7). An even better performing similarity function was the inverse of the Manhattan distance:

$$\text{sim}_i(\boldsymbol{K}, \boldsymbol{q}) = \frac{1}{\delta + \sum_j \text{abs}(\boldsymbol{K}_{ij} - \boldsymbol{q}_j)} \tag{17}$$

---

[2]The code is available at `https://github.com/HugoChateauLaurent/DND_AssociativeMemory` and is based on `https://github.com/BerenMillidge/Theory_Associative_Memory` (MIT license).

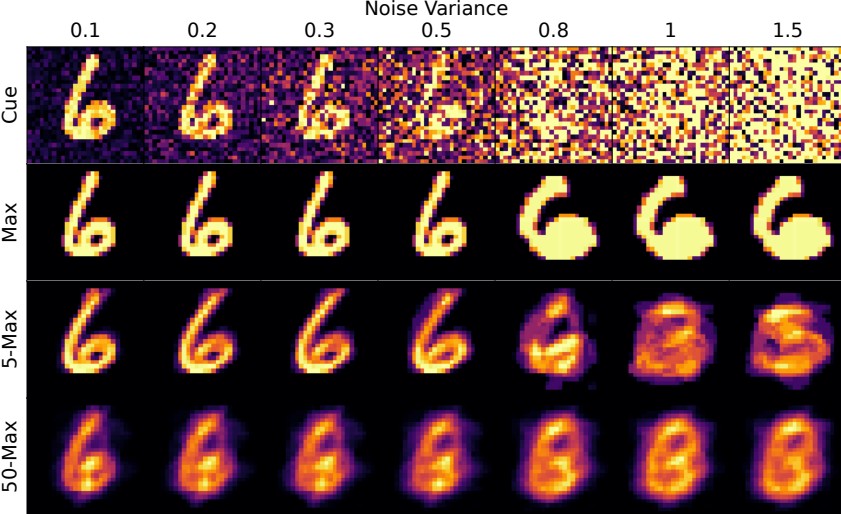

Figure 2: Example reconstructions of noisy MNIST digits by DND. The top row shows the input cue with increasing amount of noise. The following rows show the reconstruction of the stored memory using $k = 1, 5, 50$.

The capacity of the model under different similarity (Euclidean and Manhattan) and separation functions is assessed by quantifying correctly retrieved data when increasing number of MNIST, CIFAR10 and Tiny ImageNet images is stored (Figures 3 and 9 ; Table 1). Half-masked images are given as input, and a trial is correct if the sum of squared pixel differences between the output and the actual image is less than a threshold of 50. The Manhattan similarity function outperforms the Euclidean function, especially when $k$ is low. Furthermore, the best $k$ value is highly dependent on the dataset. In MNIST, the best $k$ is 5 for both Euclidean and Manhattan functions. In CIFAR10, the best $k$ is 2 with Euclidean similarity and 1 with the Manhattan function. In Tiny ImageNet, Max ($k = 1$) outperforms other functions.

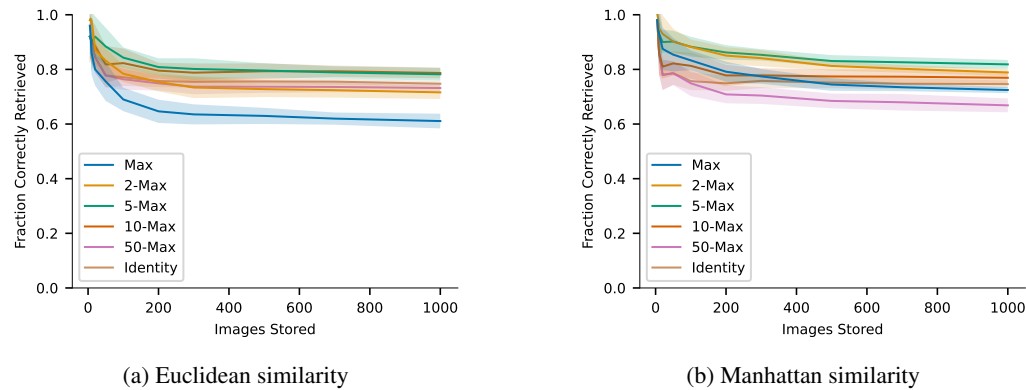

(a) Euclidean similarity

(b) Manhattan similarity

Figure 3: Capacity of associative memory with different similarity and separation functions assessed with MNIST. Plots represent the means and standard deviations of 10 simulations. A trial is correct when the difference between the output and the actual memory is under a threshold.

## 3.2 Retrieval with different functions

In order to test robustness of the memory, the ability to recall memories from noisy cues is analyzed. Independent zero-mean Gaussian noise with variance $\sigma$ is thus added to the query images pixelwise. Performance is evaluated using sets of 100 images.

Table 1: Capacity of associative memory with different similarity and separation functions assessed with MNIST, CIFAR10 and Tiny ImageNet datasets. Reported are means and standard deviations of the 10 simulations of Figure 9. For each dataset and similarity function, the best performance is highlighted in bold.

| Separator | MNIST | CIFAR10 | Tiny |
|---|---|---|---|
| Euclidean Similarity | | | |
| Max | $0.739 \pm 0.14$ | $0.220 \pm 0.18$ | $\mathbf{0.223 \pm 0.21}$ |
| 2-Max | $0.826 \pm 0.11$ | $\mathbf{0.236 \pm 0.18}$ | $0.015 \pm 0.02$ |
| 5-Max | $\mathbf{0.851 \pm 0.08}$ | $0.117 \pm 0.09$ | $0.010 \pm 0.02$ |
| 10-Max | $0.838 \pm 0.08$ | $0.095 \pm 0.08$ | $0.010 \pm 0.02$ |
| 50-Max | $0.801 \pm 0.09$ | $0.087 \pm 0.09$ | $0.010 \pm 0.02$ |
| Identity | $0.809 \pm 0.08$ | $0.088 \pm 0.09$ | $0.010 \pm 0.02$ |
| Manhattan Similarity | | | |
| Max | $0.835 \pm 0.10$ | $\mathbf{0.451 \pm 0.21}$ | $\mathbf{0.669 \pm 0.24}$ |
| 2-Max | $0.886 \pm 0.08$ | $0.369 \pm 0.20$ | $0.011 \pm 0.02$ |
| 5-Max | $\mathbf{0.887 \pm 0.07}$ | $0.106 \pm 0.08$ | $0.010 \pm 0.02$ |
| 10-Max | $0.826 \pm 0.08$ | $0.075 \pm 0.07$ | $0.010 \pm 0.02$ |
| 50-Max | $0.775 \pm 0.11$ | $0.067 \pm 0.06$ | $0.010 \pm 0.02$ |
| Identity | $0.804 \pm 0.09$ | $0.063 \pm 0.06$ | $0.010 \pm 0.02$ |

Like capacity, the best $k$ for retrieval depends on the dataset (Figure 10 Table 2). Here again, the performance is better with the Manhattan similarity than with the Euclidean similarity for low $k$, and worse for high $k$. In MNIST, the best $k$ is 50 with both Euclidean and Manhattan similarities. In CIFAR10, 2 is the best $k$. Like for capacity, Max outperforms other functions with Tiny ImageNet images.

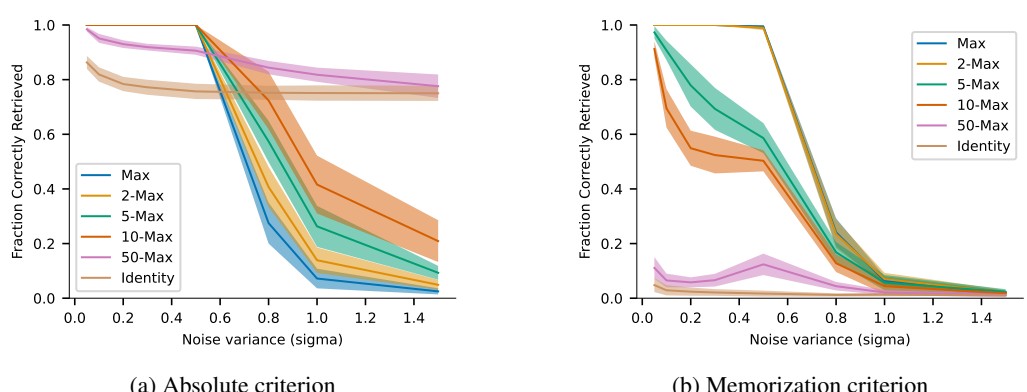

(a) Absolute criterion                    (b) Memorization criterion

Figure 4: Retrieval capability against increasing levels of noise. Plots represent the means and standard deviations of 10 simulations with different sets of MNIST images.

As hypothesized, Max is not always the best performing function with the absolute accuracy criterion, both for capacity and retrieval tasks. It can indeed be outperformed by higher $k$ values, meaning that taking into account more memories than the single most similar one can lead to more precise recall in absolute terms (i.e. as assessed by a threshold). The performance of the identity and 50-Max functions in the MNIST dataset are surprisingly good, even under very high levels of noise (Figure 4a). In fact, as pixel values are restricted to lie in the range $[0, 1]$, it is very unlikely that enough information remains in the image to correctly identify it when $\sigma > 1$. Hence, a plausible explanation is that high $k$ functions model the dataset such that they output a mixture of many images that is sometimes classified as correct retrieval, although it is not necessarily closer to the query image than

Table 2: Retrieval capability against noise. Reported are means and standard deviations of the 10 simulations of Figure 10. For each dataset and similarity function, the best performance is highlighted in bold.

| Separator | MNIST | CIFAR10 | Tiny |
|---|---|---|---|
| Euclidean Similarity | | | |
| Max | $0.667 \pm 0.44$ | $0.574 \pm 0.45$ | $\mathbf{0.580 \pm 0.44}$ |
| 2-Max | $0.692 \pm 0.41$ | $\mathbf{0.588 \pm 0.44}$ | $0.310 \pm 0.43$ |
| 5-Max | $0.735 \pm 0.37$ | $0.455 \pm 0.41$ | $0.190 \pm 0.35$ |
| 10-Max | $0.789 \pm 0.31$ | $0.357 \pm 0.39$ | $0.128 \pm 0.33$ |
| 50-Max | $\mathbf{0.904 \pm 0.09}$ | $0.205 \pm 0.30$ | $0.002 \pm 0.01$ |
| Identity | $0.830 \pm 0.10$ | $0.083 \pm 0.09$ | $0.000 \pm 0.00$ |
| Manhattan Similarity | | | |
| Max | $0.672 \pm 0.43$ | $0.627 \pm 0.44$ | $\mathbf{0.620 \pm 0.43}$ |
| 2-Max | $0.699 \pm 0.40$ | $\mathbf{0.628 \pm 0.43}$ | $0.223 \pm 0.39$ |
| 5-Max | $0.741 \pm 0.36$ | $0.424 \pm 0.38$ | $0.009 \pm 0.02$ |
| 10-Max | $0.794 \pm 0.30$ | $0.282 \pm 0.28$ | $0.002 \pm 0.01$ |
| 50-Max | $\mathbf{0.891 \pm 0.07}$ | $0.085 \pm 0.06$ | $0.000 \pm 0.00$ |
| Identity | $0.781 \pm 0.05$ | $0.049 \pm 0.02$ | $0.000 \pm 0.00$ |

any other of the dataset. This is especially true for MNIST which contains simple pictures that are more similar to each other than CIFAR10 and Tiny ImageNet.

### 3.3 Performance with memorization criterion

In order to prevent associative memory models from modeling the statistics of the dataset rather than focusing on the query image to output the actual memory, a novel criterion is introduced. Instead of the absolute threshold, retrieval must be good relatively to other images. More precisely, the novel criterion is such that a trial is correct if and only if the sum of squared pixel differences between the truth and the output is lower or equal to the sum of squared pixel differences between the output and any other memory, that is, if:

$$\sum_j (\boldsymbol{z_j} - \boldsymbol{K}_{cj})^2 = \min_i \sum_j (\boldsymbol{z_j} - \boldsymbol{K}_{ij})^2 \tag{18}$$

where $\boldsymbol{K}_c$ is the correct pattern to retrieve.

Capacity is now assessed using this new criterion (Figures 5 and 11 ; Table 3). The Manhattan function still outperforms the Euclidean similarity. Most crucially, the best performance is always obtained with the Max function.

Retrieval is then tested with the new criterion (Figures 4b and 12 ; Table 4). Once again, the best performance is obtained with the Manhattan similarity. Furthermore, $k = 1$ almost always outperforms other values. Note that performance with $k = 2$ is very similar.

### 3.4 Relationship between $k$-Max and Softmax

Like $k$-Max, the Softmax function virtually cancels out the contribution of distant memories, especially when $\beta$, the scaling parameter of its input, is high. It does it by normalizing exponentiated similarity scores:

$$\text{Softmax}(\boldsymbol{x}) = \frac{e^{\beta \boldsymbol{x}}}{\sum_i e^{\beta \boldsymbol{x}_i}} \tag{19}$$

While $k$ is a discrete parameter, $\beta$ is continuous, which makes the Softmax function harder to optimize but perhaps more flexible. Here, the two separation functions are compared. For each dataset, 100 images are encoded. The noise is set to 1 for MNIST and 0.75 for CIFAR10 dataset and Tiny ImageNet. The results are shown in Figures 6a and 13 with the absolute criterion and in Figures 6b

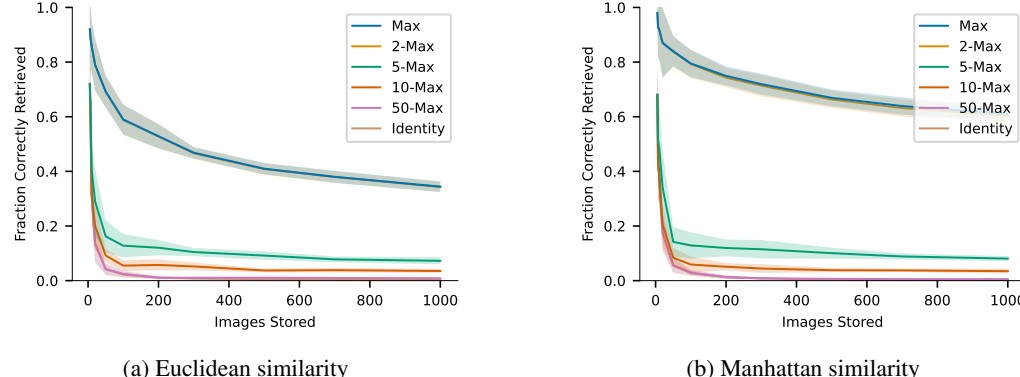

(a) Euclidean similarity          (b) Manhattan similarity

Figure 5: Capacity of associative memory with different similarity and separation functions assessed with MNIST. Plots represent the means and standard deviations of 10 simulations. Here, a trial is correct when the difference between the output and the actual memory is lower than the difference between the output and any other stored memory.

Table 3: Capacity of associative memory with different similarity and separation functions assessed with MNIST, CIFAR10 and Tiny ImageNet datasets. Reported are means and standard deviations of the 10 simulations of Figure 11. For each dataset and similarity function, the best performance is highlighted in bold.

| Separator | MNIST | CIFAR10 | Tiny |
|---|---|---|---|
| Euclidean Similarity | | | |
| Max | $\mathbf{0.624 \pm 0.22}$ | $\mathbf{0.223 \pm 0.19}$ | $\mathbf{0.223 \pm 0.21}$ |
| 2-Max | $\mathbf{0.624 \pm 0.22}$ | $0.222 \pm 0.19$ | $0.222 \pm 0.21$ |
| 5-Max | $0.256 \pm 0.24$ | $0.132 \pm 0.13$ | $0.112 \pm 0.14$ |
| 10-Max | $0.199 \pm 0.24$ | $0.104 \pm 0.14$ | $0.098 \pm 0.13$ |
| 50-Max | $0.171 \pm 0.25$ | $0.086 \pm 0.14$ | $0.095 \pm 0.14$ |
| Identity | $0.169 \pm 0.25$ | $0.085 \pm 0.14$ | $0.095 \pm 0.14$ |
| Manhattan Similarity | | | |
| Max | $\mathbf{0.793 \pm 0.14}$ | $\mathbf{0.502 \pm 0.24}$ | $\mathbf{0.669 \pm 0.24}$ |
| 2-Max | $0.790 \pm 0.14$ | $0.486 \pm 0.25$ | $0.505 \pm 0.36$ |
| 5-Max | $0.255 \pm 0.22$ | $0.176 \pm 0.19$ | $0.155 \pm 0.20$ |
| 10-Max | $0.187 \pm 0.22$ | $0.103 \pm 0.15$ | $0.119 \pm 0.18$ |
| 50-Max | $0.163 \pm 0.23$ | $0.090 \pm 0.15$ | $0.113 \pm 0.18$ |
| Identity | $0.162 \pm 0.23$ | $0.088 \pm 0.15$ | $0.113 \pm 0.18$ |

and 14 with the memorization criterion. Except for the condition with CIFAR10, Euclidean similarity and absolute criterion (Figure 13b), the Softmax can always outperform $k$-Max.

## 4 Discussion

In this paper, DND, which has initially been introduced in the context of reinforcement learning (6), has been shown to be mathematically related to Hopfield Networks (1). The Universal Hopfield Network framework has recently been introduced to encompass the traditional Hopfield Network, modern variants and related models (7). These models recall memories with a common sequence of operations: similarity, separation and projection. It has been shown that retrieval from a DND is also done with these operations. Hence, a DND is an instance of the Universal Hopfield Network framework. For the sake of mathematical analysis, a continuous approximation of DND recall has

Table 4: Retrieval capability against noise. Reported are means and standard deviations of the 10 simulations of Figure 12. For each dataset and similarity function, the best performance is highlighted in bold.

| Separator | MNIST | CIFAR10 | Tiny |
|---|---|---|---|
| | Euclidean Similarity | | |
| Max | **0.661 ± 0.44** | 0.574 ± 0.45 | **0.580 ± 0.44** |
| 2-Max | **0.661 ± 0.44** | **0.575 ± 0.44** | 0.579 ± 0.44 |
| 5-Max | 0.576 ± 0.41 | 0.366 ± 0.42 | 0.416 ± 0.45 |
| 10-Max | 0.522 ± 0.39 | 0.288 ± 0.40 | 0.314 ± 0.41 |
| 50-Max | 0.246 ± 0.34 | 0.134 ± 0.29 | 0.160 ± 0.32 |
| Identity | 0.150 ± 0.32 | 0.029 ± 0.05 | 0.101 ± 0.24 |
| | Manhattan Similarity | | |
| Max | **0.665 ± 0.44** | **0.621 ± 0.44** | **0.622 ± 0.43** |
| 2-Max | 0.664 ± 0.43 | **0.621 ± 0.44** | **0.622 ± 0.43** |
| 5-Max | 0.523 ± 0.36 | 0.316 ± 0.37 | 0.352 ± 0.40 |
| 10-Max | 0.422 ± 0.31 | 0.177 ± 0.25 | 0.228 ± 0.33 |
| 50-Max | 0.063 ± 0.04 | 0.026 ± 0.02 | 0.030 ± 0.03 |
| Identity | 0.022 ± 0.02 | 0.012 ± 0.00 | 0.012 ± 0.01 |

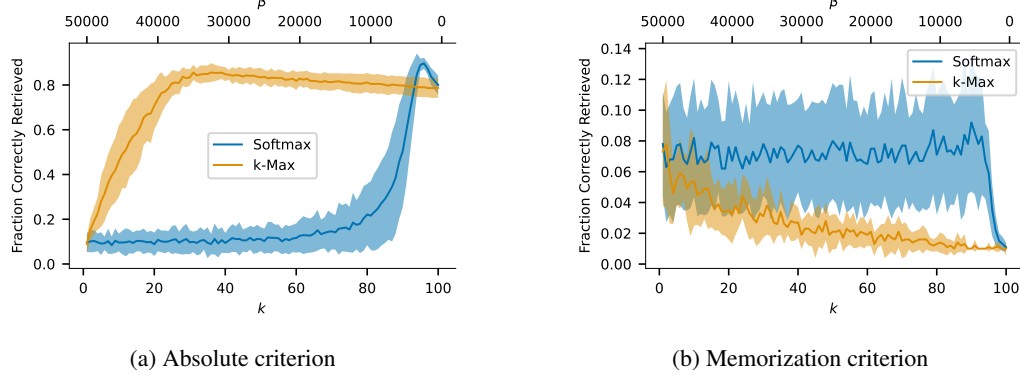

(a) Absolute criterion

(b) Memorization criterion

Figure 6: Retrieval capability as a function of $k$ and $\beta$ parameters of the $k$-Max and Softmax separation functions respectively. In each simulation, 100 MNIST images are encoded, then queried with a noise of 1. Plots represent the means and standard deviations of 10 simulations with different sets of images.

been proposed to comply with the requirements of the energy function of UHN and derive a second Lyapunov function of the dynamics.

This novel link connects the fields of associative memory and reinforcement learning. The similarity function of DND is Euclidean and has already been shown to yield high capacity (7). On the other hand, the $k$-nearest neighbor is not commonly used as a separation function for associative memory. One thing to note is that the time complexity of $k$-nearest neighbor search is $\mathcal{O}(\log n)$ when implemented with k-d trees (11). In contrast, the Softmax function has a time complexity of O(n). Thus, one of the present objectives was to assess the performance of the more efficient $k$-Max separation function.

Interestingly, $k$ controls the degree of separation, and setting $k = 1$ is equivalent to using the Max function studied by (7). While having theoretically unbounded capacity, the Max function can transition sharply from one memory attractor to another when noise of increasing amplitude is added to the query. With Figure 2, it was hypothesized that higher $k$ values could be better at modelling

datasets and improve the performance assessed in absolute terms. Simulations indeed revealed higher capacity and better retrieval from noisy queries with $k > 1$, especially with simple datasets like MNIST. However, these results depend on the way performance is evaluated. Traditionally, the evaluation of retrieval is based on some distance evaluation of the memory output and the actual image, which must not exceed some threshold fixed by the experimenter. This is a widespread method for evaluating associative memory models, but one must choose the threshold wisely, as setting it too high can result in false positives with the model grossly reproducing statistics of the dataset (generalization). This for example seems to be the case when retrieval is assessed using the MNIST dataset. The performance of 50-Max (and even the identity function) remains high despite very strong noise. Therefore, another way of evaluating retrieval was introduced, which does not consider the output in absolute terms, but rather compares it to the whole memory set. Retrieval is deemed correct if and only if the output resembles the actual image more than any other stored memory. Memory models thus cannot benefit from modeling statistics of the dataset, and must rather focus on recalling the distinguishing characteristics of the query (memorization). Using this method, the Max function outperforms the others. Ideally, performance should be evaluated in both absolute and relative terms to ensure that recall is accurate and stands out from other memories.

This raises the question of what is the function of associative memory. Modeling statistics of a dataset is related to generalization, which is typically the main goal of machine learning. The objective of associative memory is somewhat different. Instead of generalizing, an associative memory aims to recall the exact information corresponding to the individual memory. This is reminiscent of the division of labor between episodic and semantic memory (12). When it comes to episodic control however, that is the use of episodic memories for action control, some generalization is desirable. This is especially the case in Neural Episodic Control in which the selection of actions only relies on episodes, the DND thus constituting a bottleneck. Initially, episodic control (not to be confused with its implementation in Neural Episodic Control) has been introduced as a way of speeding up the learning of reinforcement agent and, after the initial episodic control phase, it is desirable that more robust controllers can take over (5). A biologically inspired alternative to Neural Episodic Control would be to supplement episodic memory with other controllers whose function is to generalize. The episodic memory would then no longer be a bottleneck, and could instead be devoted to memorizing the specifics of situations. That being said, there is also an ongoing debate about the fact that episodic memory could also integrate a part of generalization and not only store the specificities of episodes (13; 14; 15).

## 5    Limitations and Future Work

In this paper, we mainly focused on evaluating the capacity and retrieval performance of associative memory models. Conversely to the application of DND to associative memory, the novel theoretical link also implies that any instance of UHN can be used for episodic control. It is possible that the Manhattan function could consistently improve sample efficiency, outperforming the Euclidean kernel of DND, as it does on the associative memory tasks. The Softmax function, which has been proven powerful in transformers (16), and more performant than $k$-Max in the present study, could also improve episodic control agents. RL experiments are being conducted in this regard.

Finally, the fact that DND is theoretically related to Hopfield Networks provides a biological basis to Neural Episodic Control, as the most influential model of the hippocampus relies heavily on similar associative memory mechanisms (17). Hence, this study opens up new avenues of research at the frontier of the fields of associative memory, reinforcement learning and neuroscience.

## Acknowledgements

Experiments presented in this paper were carried out using the PlaFRIM experimental testbed, supported by Inria, CNRS (LABRI and IMB), Université de Bordeaux, Bordeaux INP and Conseil Régional d'Aquitaine (see `https://www.plafrim.fr`).

Both authors were funded by Inria. No additional sources of funding were received in support of this work.

The authors declare that they have no competing financial interests or relationships with entities that could be perceived to influence the work presented in this paper.

The authors would like to thank Thierry Viéville for his help deriving the energy functions, as well as Dolton Fernandes, who discovered the Universal Hopfield Network paper.

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

# A    Convergence of the threshold $\Theta$

From Equations 13 and 15 we obtain:

$$\Theta^{t+1} = F\left(\Theta^t\right), \tag{20}$$

$$F\left(\Theta\right) = \Theta + \alpha \underbrace{\left[ -k + \sum_{i=1}^{n} \sigma\left[ \beta_k\left( \sigma\left(\beta(\boldsymbol{x}_i - \Theta)\right) - \frac{1}{2} \right) \right] \right]}_{\Delta(\Theta)}, \tag{21}$$

thus

$$F'\left(\Theta\right) = 1 - \alpha \underbrace{\sum_{i=1}^{n} \sigma'\left( \beta_k\left( \sigma\left(\beta(\boldsymbol{x}_i - \Theta)\right) - \frac{1}{2} \right) \right) \beta_k\, \sigma'\left(\beta(\boldsymbol{x}_i - \Theta)\right) \beta}_{\delta(\Theta) = -\Delta'(\Theta)} \tag{22}$$

Since from Equation 12 of the Sigmoid: $0 < \sigma'\left(x\right) = \frac{1}{\left(e^{x/2} + e^{-x/2}\right)^2} \leq 1/4$, while by design $0 < \alpha$, $0 < \beta$, $0 < \beta_k$, we obtain:

$$F'\left(\Theta\right) = 1 - \delta\left(\Theta\right) \text{ with } 0 < \delta\left(\Theta\right) \leq \frac{\alpha\, n\, \beta_k\, \beta}{16} \tag{23}$$

so that if $\alpha < \frac{16}{n\,\beta_k\,\beta}$ then $0 < \delta\left(\Theta\right) < 1$ thus $0 < F'\left(\Theta\right) < 1$ so that the recurrent series defining $\Theta_\infty$ is monotonic convergent[3]. This convergence is verified for all $\boldsymbol{x}$, so that if their values vary during the convergence, the final value of $\Theta$ may vary, but always in converging mode.

Furthermore, $\Delta\left(\Theta\right)$ decreases along the iteration and can be used as energy (i.e. Lyapounov function) for the recurrence, in complement of the abstract energy given in Equation 11. To avoid any interference between both converging processes, at the implementation level, the continuous value of $\Theta$ is calculated as a fast local iteration loop, so that it is then almost constant when adjusting the dynamic related to Equation 11.

---

[3]For $\alpha < \frac{32}{k\,\beta_k\,\beta}$ we still have $\left|F'\left(\Theta\right)\right| < 1$, since $-1 < F'\left(\Theta\right) < 1$, thus convergence, but the convergence may be oscillatory. Since $\lim_{t\to\infty}\left|F'\left(\Theta\right)\right| = 1$ the fixed point is at the edge of stability, thus monotonic convergence is preferable at the numerical level.

# B Example reconstructions

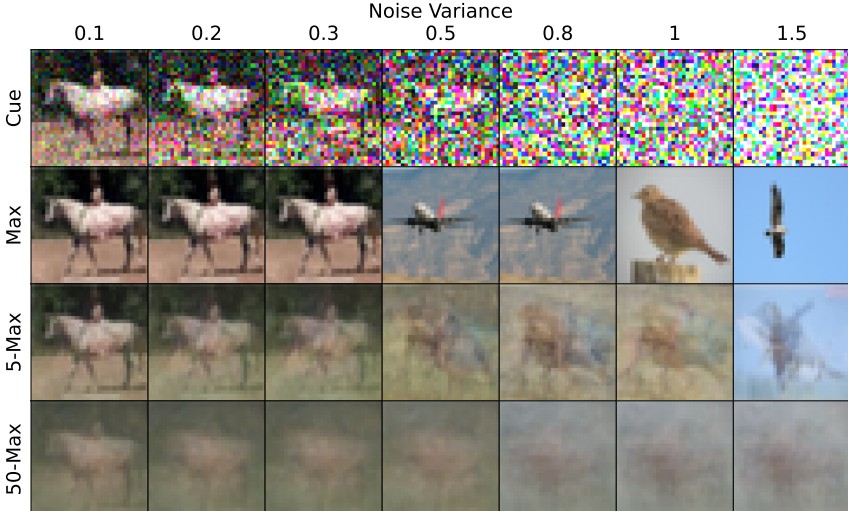

Figure 7: Example reconstructions of noisy CIFAR10 images by DND. The top row shows the input cue with increasing amount of noise. The following rows show the reconstruction of the stored memory using $k = 1, 5, 50$.

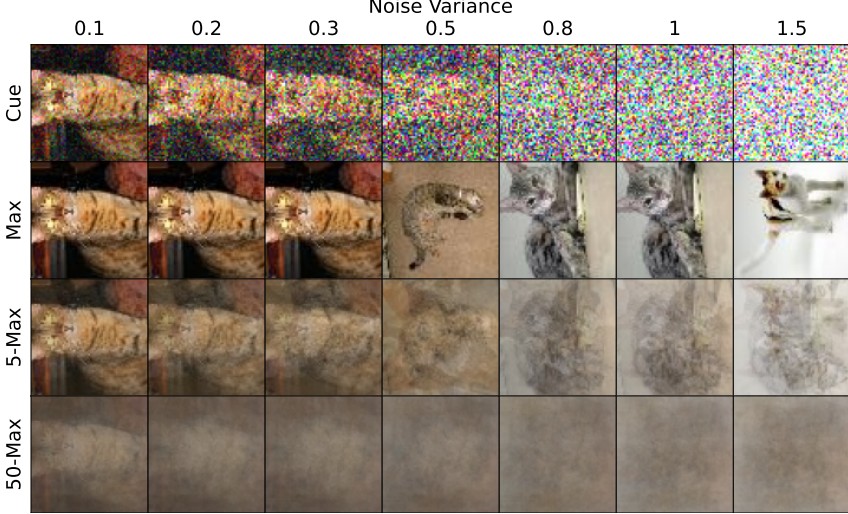

Figure 8: Example reconstructions of noisy Tiny ImageNet images by DND. The top row shows the input cue with increasing amount of noise. The following rows show the reconstruction of the stored memory using $k = 1, 5, 50$.

# C Capacity with absolute criterion

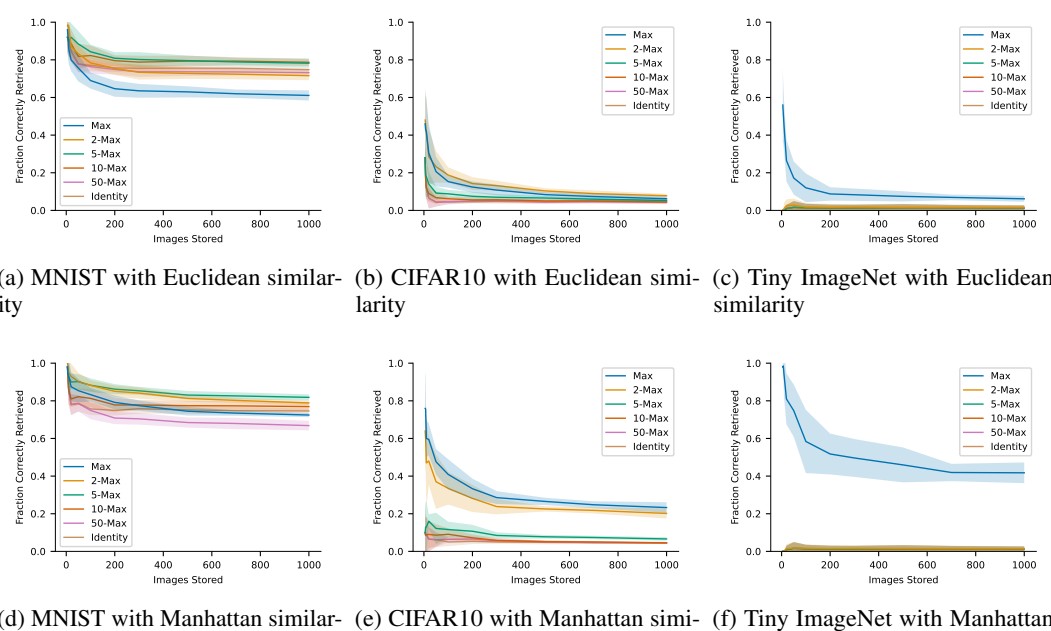

(a) MNIST with Euclidean similarity

(b) CIFAR10 with Euclidean similarity

(c) Tiny ImageNet with Euclidean similarity

(d) MNIST with Manhattan similarity

(e) CIFAR10 with Manhattan similarity

(f) Tiny ImageNet with Manhattan similarity

Figure 9: Capacity of associative memory with different similarity and separation functions assessed with MNIST, CIFAR10 and Tiny ImageNet datasets. Plots represent the means and standard deviations of 10 simulations. A trial is correct when the difference between the output and the actual memory is under a threshold.

# D Retrieval with absolute criterion

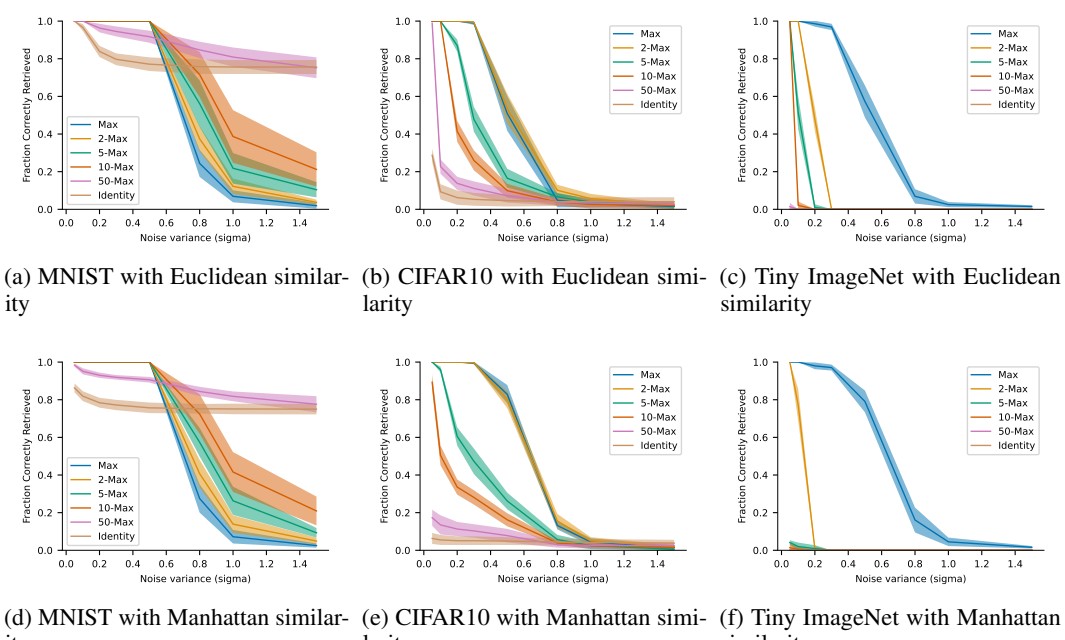

(a) MNIST with Euclidean similarity

(b) CIFAR10 with Euclidean similarity

(c) Tiny ImageNet with Euclidean similarity

(d) MNIST with Manhattan similarity

(e) CIFAR10 with Manhattan similarity

(f) Tiny ImageNet with Manhattan similarity

Figure 10: Retrieval capability against increasing levels of noise. Plots represent the means and standard deviations of 10 simulations with different sets of images. A trial is correct when the difference between the output and the actual memory is under a threshold.

# E    Capacity with memorization criterion

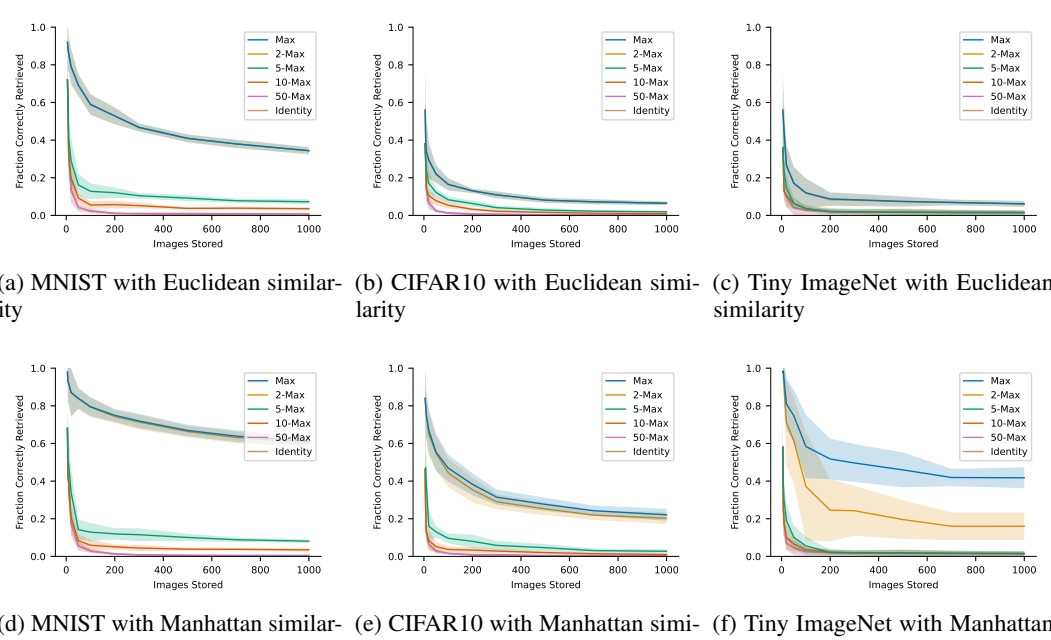

(a) MNIST with Euclidean similarity

(b) CIFAR10 with Euclidean similarity

(c) Tiny ImageNet with Euclidean similarity

(d) MNIST with Manhattan similarity

(e) CIFAR10 with Manhattan similarity

(f) Tiny ImageNet with Manhattan similarity

Figure 11: Capacity of associative memory with different similarity and separation functions assessed with MNIST, CIFAR10 and Tiny ImageNet datasets. Plots represent the means and standard deviations of 10 simulations. Here, a trial is correct when the difference between the output and the actual memory is lower than the difference between the output and any other stored memory.

# F    Retrieval with memorization criterion

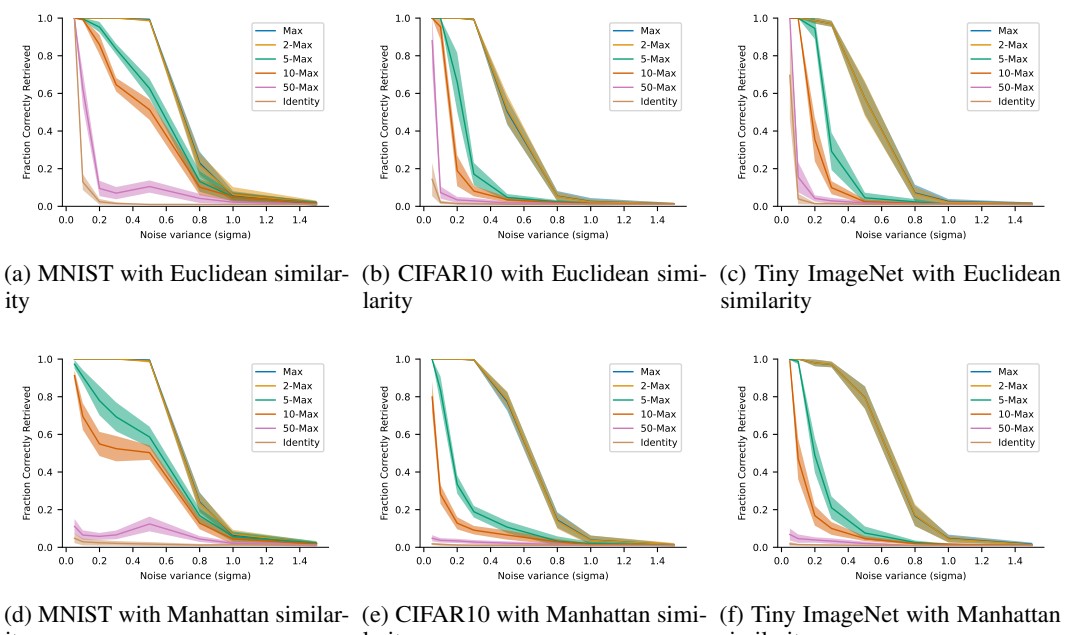

(a) MNIST with Euclidean similarity

(b) CIFAR10 with Euclidean similarity

(c) Tiny ImageNet with Euclidean similarity

(d) MNIST with Manhattan similarity

(e) CIFAR10 with Manhattan similarity

(f) Tiny ImageNet with Manhattan similarity

Figure 12: Retrieval capability against increasing levels of noise. Plots represent the means and standard deviations of 10 simulations with different sets of images. Here, a trial is correct when the difference between the output and the actual memory is lower than the difference between the output and any other stored memory.

# G  $\beta$ vs. $k$ with absolute criterion

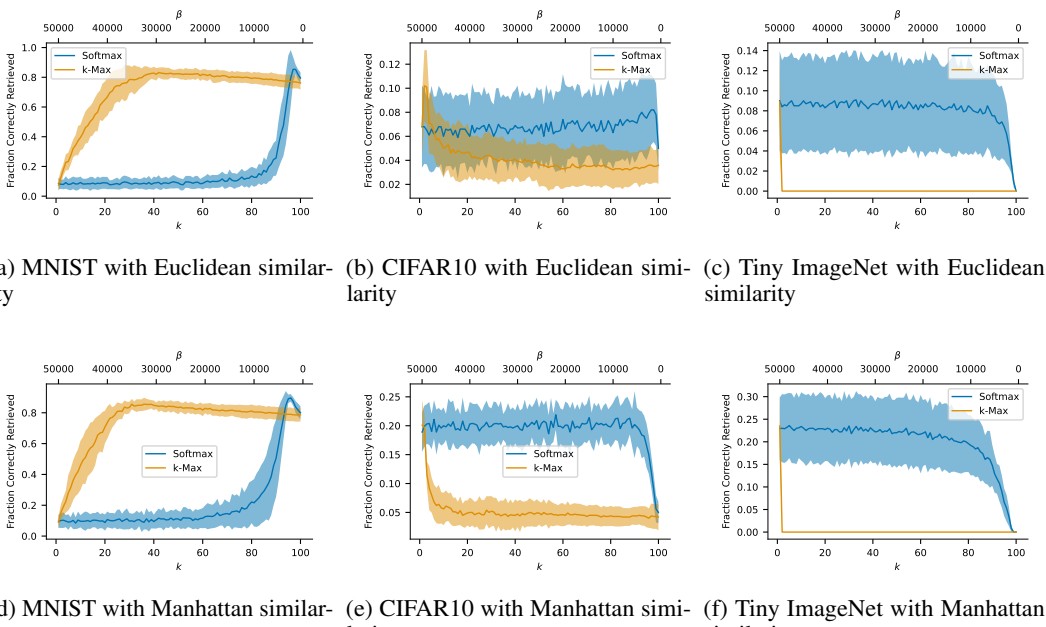

(a) MNIST with Euclidean similarity

(b) CIFAR10 with Euclidean similarity

(c) Tiny ImageNet with Euclidean similarity

(d) MNIST with Manhattan similarity

(e) CIFAR10 with Manhattan similarity

(f) Tiny ImageNet with Manhattan similarity

Figure 13: Retrieval capability as a function of $k$ and $\beta$ parameters of the $k$-Max and Softmax separation functions respectively. Plots represent the means and standard deviations of 10 simulations with different sets of images. A trial is correct when the difference between the output and the actual memory is under a threshold.

# H  $\beta$ vs. $k$ with memorization criterion

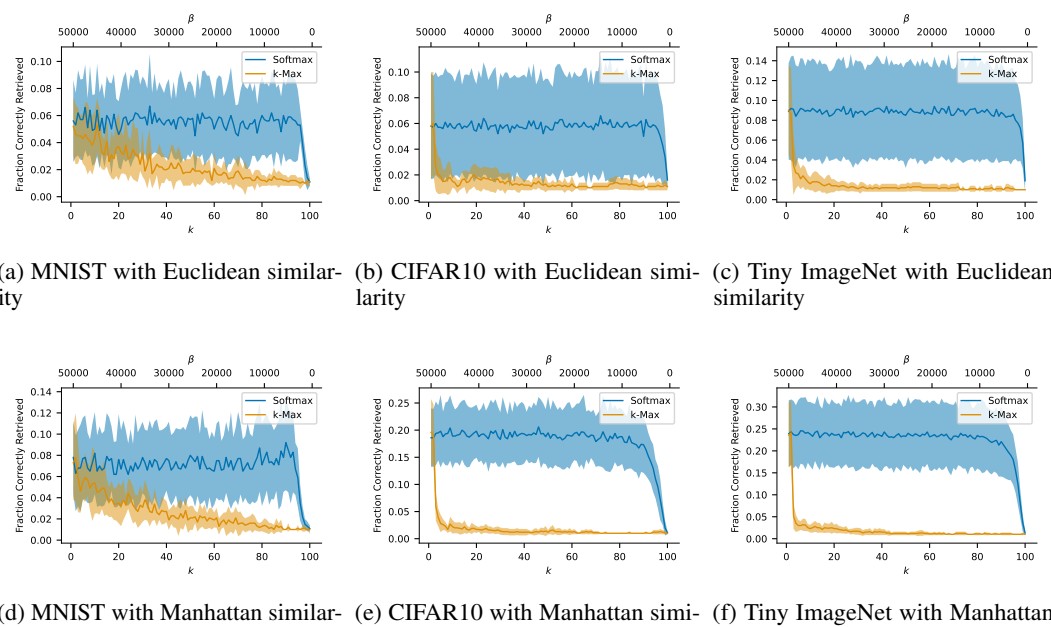

(a) MNIST with Euclidean similarity

(b) CIFAR10 with Euclidean similarity

(c) Tiny ImageNet with Euclidean similarity

(d) MNIST with Manhattan similarity

(e) CIFAR10 with Manhattan similarity

(f) Tiny ImageNet with Manhattan similarity

Figure 14: Retrieval capability as a function of $k$ and $\beta$ parameters of the $k$-Max and softmax separation functions respectively. Plots represent the means and standard deviations of 10 simulations with different sets of images. Here, a trial is correct when the difference between the output and the actual memory is lower than the difference between the output and any other stored memory.

