# OpenReview forum: "Relating Hopfield Networks to Episodic Control"
_NeurIPS.cc/2024/Conference — NeurIPS 2024 poster_

### Official Review · Reviewer_w6Cb · 2024-07-09

**Soundness:** 2
**Presentation:** 2
**Contribution:** 3
**Rating:** 6
**Confidence:** 4

**Summary:**

The paper shows that the Differentiable Neural Dictionary (DND) used in the context of reinforcement learning is mathematically equivalent to an Hopfield network with heteroassociative memories. Based on this observation, the paper generalises DND using the formulation of Universal Hopfield Networks (UHN) and studies the effect on 'retrieval' and 'capacity' of different separation and similarity functions. Additionally, the authors introduce a new criterion to measure the retrieval capability of an associative memory, which, they argue is a better proxy for memorisation.

**Strengths:**

The paper is generally well written and has a good structure. Particular strengths reside in:

- Solid contribution: the characterisation of differentiable neural dictionaries as associative memories can help future research in the field of neural episodic control by applying the advancements of the research on associative memories to the DND implementation.
- Soundness of the approach: the authors establish the connection theoretically and then experimentally study the effect of changing the form of the DND associative memory on the memory capacity. They use enough baselines and different datasets, so the results are convincing.

**Weaknesses:**

I think the paper is relatively light, meaning that there is little content. In particular, the authors focus most of the experiments on studying the effect of different similarity and separation functions on capacity for storing images, whilst they could have investigated the implications in the reinforcement learning domain more. It's unclear why the study of the 'max' separating function in comparison to softer versions thereof is so important for this paper: it indeed seems that section 3 belongs to another paper, which talks about the characteristics of different separating functions.
Another example is the Discussion section, which occupies a page and a half mostly with speculative sentences and arbitrary connections to neuroscience research, which are far from justified in the context of this paper. It would have been much better if the paper actually empirically or theoretically attempted at establishing this connections. For an example see lines 217-219.

**Questions:**

Some minor points and questions:
- In Section 2 what are the dimensions of the vectors and matrices, e.g. V, K, q, $\phi$, etc? Please rectify directly in the paper.
- The definition of the similarity function used by DND has an unusual $\delta$ in it. That is probably there for numerical stability reasons and should be ignored in the mathematical treatment.
- Equations (7-9) are obvious and not needed.
- The dynamic and differentiable way of defining the threshold (Equations 15-16) is clever but not necessary since you can just zero-out the non top-K ones? (which you do in the experiments if I understood correctly)
- It is not really clear to me the difference between capacity and retrieval, which is central to section 3 so you might want to state it somewhere at the beginning of it. The line between them seems very thin to me.
- Line 134-135 maybe you wanted to say the opposite?
-In section 3 you are testing an hypothesis which has not been clearly spelled out before. If I understand correctly, your hypothesis is that the k-max function is useful for generalisation or memorisation depending on the value of k chosen. I believe you have tried to mention this in the introduction of Section 3 (lines 109-112) but I think it's not clear and confusing at the moment.

**Limitations:**

I can see no major limitations, but the paper could have been developed more.

---

> ### Author Rebuttal · Authors · 2024-08-02
>
> We thank Reviewer w6Cb for the constructive feedback and for acknowledging the strengths of our paper, including the solid contribution and soundness of our approach. We appreciate the recognition of our work in characterizing Differentiable Neural Dictionaries (DNDs) as associative memories and its potential to advance research in neural episodic control. We also thank you for pointing raising valid comments. We directly included explicitly changes you proposed (e.g. clarification of matrix dimensionality), and we address additional concerns and questions that call for a response below.
>
> 1. Comment: "I think the paper is relatively light [...]"
>     - Answer: We agree that further exploring reinforcement learning (RL) implications is valuable, and we are actively pursuing this direction in ongoing work. The primary focus of this manuscript was to describe the theoretical connection between DNDs and Hopfield networks and to evaluate the memory capabilities of DNDs. Due to the page limit and dense supplementary information, we concentrated on establishing this foundational work. We look forward to sharing new results on RL applications in future publications.
>
> 2. Comment: "It's unclear why the study of the 'max' separating function in comparison to softer versions thereof is so important for this paper: it indeed seems that section 3 belongs to another paper, which talks about the characteristics of different separating functions."
>     - Answer: Thank you for your feedback. All experiments, including the one you mentioned, evaluate the k-max (not only max) function to demonstrate its unique properties and effectiveness in different contexts. We believe this comparison provides valuable insights into optimizing memory retrieval strategies. More precisely, it suggests that NEC could be improved by replacing k-max by softmax. Does this clarification address your concern?
>
> 3. Comment: "Another example is the Discussion section, which occupies a page and a half mostly with speculative sentences and arbitrary connections to neuroscience research"
>     - Answer: Thank you for pointing that out. We do think the connection to neuroscience can be made and "adds an interesting interdisciplinary perspective" (Reviewer 4Gfy). However, we replaced speculative content by referenced claims.
>
> 4. Question: "The dynamic and differentiable way of defining the threshold (Equations 15-16) is clever but not necessary since you can just zero-out the non top-K ones? (which you do in the experiments if I understood correctly)"
>     - Answer: You are right. This new definition was introduced to derive the energy functions for neural episodic control, but it was not used in the experiments.
>
> 5. Question: "It is not really clear to me the difference between capacity and retrieval, which is central to section 3 so you might want to state it somewhere at the beginning of it. The line between them seems very thin to me."
>     - Answer: Thank you for bringing this to our attention. We adhered to the distinction of Millidge et al. (2022). Capacity refers to the maximum number of images the DND can store while maintaining accurate memory representation. Retrieval, on the other hand, measures the model’s ability to accurately recall stored images when subjected to increasing levels of noise. We have added a definition and explanation of these terms in the manuscript to clarify their distinction.
>
> We appreciate the constructive feedback provided by Reviewer w6Cb, which has been instrumental in enhancing the clarity and depth of our manuscript. We have addressed the concerns and questions raised, and we believe these revisions have strengthened our work. We are committed to further exploring the implications of our findings in reinforcement learning and look forward to sharing these insights in future publications. Thank you again for your thoughtful and valuable comments, which have helped us improve our contribution to the field.

---

### Official Review · Reviewer_ebPC · 2024-07-11

**Soundness:** 4
**Presentation:** 2
**Contribution:** 2
**Rating:** 6
**Confidence:** 4

**Summary:**

This paper introduces a formulation of an energy function which involves retrieving top-k memories while making a connection to both Neural Episodic Control and Associative Memory. The novel energy function utilized in this work demonstrates superior performance in the retrieval setting which involves image in-painting where the top half of an image is masked out. Additionally, the work explores the efficacy of the model under the retrieval setting in which queries are perturbed with a range of noise values.

**Strengths:**

The proposed energy function is novel and well thought. The formulation follows the Neural Episodic Control and Universal Hopfield Network (UHN) paradigms. Moreover, the experiments are interesting, and the model is shown to have great performance, specifically the illustrations of improved memorization capacity across a variety of k-Max functions demonstrated on MNIST. When it comes to denoising, the proposed model is able to recover patterns very well as k increases.

**Weaknesses:**

Although the experiments are good, many of the experiments are ablation studies of the introduced energy function, while there is one experiment section contrasting the proposed energy function and softmax based energy function, which illustrates a worse performance of the new function. Moreover, when dealing with RGB images, the function performs badly as k-neighbors increases. Finally, the connection to RL is fascinating but it seems too brief in the paper.

**Questions:**

Why is the range of $\beta$, 0 to 50000, chosen for figure 5? Such values seem to be too large, while the range of k makes sense.

Looking at figures 6 and 8, it seems that when k = 1, the model performs best on CIFAR10 and Tiny ImageNet which is an opposite trend to MNIST. What is the explanation behind this fact?

For figure 7, as k increases, the performance of the model increases given the setting of denoising. What is the number of images that the model is being evaluated on? Could a visual example of denoising Tiny ImageNet images be provided?

What is the threshold utilize to determine whether a recovered pattern is memorized or not for each of the dataset?

"The novel criterion is such that a trial is correct if and only if the squared pixel difference between the truth and the output is lower than it is between the output and any other memory" --- Why is there no equation which describes this important criterion?

General comment: Since the maximum chosen k value is set to 50 in the tables, I think it would be good to include k = 100 in such tables as it is demonstrated in figure 5.

**Limitations:**

The connection to Neural Episodic Control and RL is too brief. Additionally, the function does not beat the softmax based energy function in terms of retrieval while as the number of neighbors k increases, the performance is worsen on CIFAR10 and TinyImageNet datasets.

---

> ### Author Rebuttal · Authors · 2024-08-02
>
> [concise because hitting char lim]
>
> Thank you for your constructive feedback and insights on our work.
>
> 1. Comment: "Although the experiments are good, many of the experiments are ablation studies of the introduced energy function, while there is one experiment section contrasting the proposed energy function and softmax based energy function, which illustrates a worse performance of the new function."
>     - Answer: Yes, and we think these are important findings. The Manhattan similarity function consistently outperforms the Euclidean one, even with softmax, as noted by Millidge et al. (2022). This suggests that NEC can be improved through our connection between the DND and UHN. We also argue that softmax could improve the flexibility and performance of NEC. Conversely, k-max constitutes a strong alternative due to its superior search complexity, especially when implemented with a k-d tree, as we now argue in the new version of the manuscript.
>
> 2. Comment: "Moreover, when dealing with RGB, the function performs badly as k increases."
>     - Answer: Several factors could indeed influence how robust performance is to changes in k, such as the use of RGB images, the higher dimensionality and more naturalistic nature of complex images compared to MNIST.
>
> 3. Comment: "The connection to RL is fascinating but it seems too brief in the paper."
>     - Answer: Thank you for finding the connection fascinating. Our paper establishes a comprehensive theoretical link between NEC and associative memory models. Additionally, we present a set of new empirical results demonstrating the applicability of NEC to associative memory tasks. We are now working on exploring how insights from associative memory can in turn enhance NEC.
>
> 4. Question: "Why is the range of beta, 0 to 50000, chosen for figure 5? Such values seem to be too large, while the range of k makes sense."
>     - Answer: The experiment for Figure 5 is computationally expensive, so we aimed to identify peak performance across a wide range for all datasets and similarity functions. We found that peak performance does not always occur with low beta values (fig. 11, 10). Finally, the range already demonstrates that softmax most often outperforms k-max. Please let us know if you have any more concerns.
>
> 5. Question: "Looking at figures 6 and 8, it seems that when k = 1, the model performs best on CIFAR10 and Tiny ImageNet which is an opposite trend to MNIST. What is the explanation behind this fact?"
>     - Answer: We believe your comment may be a reference to Figures 6 and 7 instead. We indeed observe that for lower-dimensional datasets like MNIST (and CIFAR-10 to a lesser extent), values of k>1 can lead to improved performance. This finding contradicts the statement by Millidge et al. (2022) that k=1 is always optimal and motivates our introduction of a new performance criterion where k=1 becomes optimal. These points are discussed in our abstract (lines 8 to 14) and throughout the manuscript.
>
> 6. Question: "For figure 7, as k increases, the performance of the model increases given the setting of denoising. What is the number of images that the model is being evaluated on? Could a visual example of denoising Tiny ImageNet images be provided?"
>     - Answer: Thank you very much for highlighting this crucial detail. We apologize for the oversight. Similar to the approach used by Millidge et al. (2022), we evaluated the model using sets of 100 images. We have now included this information in the revised manuscript. We also appreciate your suggestion to provide a visual example of denoising, and have added such examples for Tiny ImageNet in the supplementary material.
>
> 7. Question: "What is the threshold utilize to determine whether a recovered pattern is memorized or not for each of the dataset?"
>     - Answer: The error threshold is used for the absolute criterion and is set to 50, consistent with the approach used by Millidge et al. (2022), who utilized the same datasets. We mentioned this criterion in line 125 of the manuscript, and we have now made it clearer in the revised version to ensure this important detail is easily accessible.
>
> 8. Question: "Why is there no equation which describes [the generalization] criterion?"
>     - Answer: Thank you for pointing this out. Upon review, we realized that our previous description was incorrect. The correct criterion allows the sum of squared pixel differences to be equal, not just less, for a trial to be correct. We appreciate your suggestion to include an equation, as it prompted us to clarify this aspect.
>
>         The retrieval output is denoted as $z$, and the matrix of stored memories is $K$. A trial is considered correct if the sum of squared pixel differences between $z$ and the correct memory $K_{\text{correct}}$ is equal to the minimum difference between $z$ and any memory $K_i$, i.e. $z - K_{\text{correct}} \|^2 = \min_{i} \| z - K_i \|^2$.
>         This adjustment to the phrasing does not affect our results, as our implementation already adhered to this correct criterion. We have included this equation and corrected the phrasing in the revised manuscript to ensure clarity and accuracy.
>
> 9. Comment: "Since the maximum chosen k value is set to 50 in the tables, I think it would be good to include k = 100 in such tables as it is demonstrated in figure 5."
>     - Answer: Thank you for your suggestion regarding the inclusion of k=100 in the tables, but we do not have such data. For tables, the max is set to k=50 like in the original NEC paper. Other experiments have consistently shown that performance peaks before k=50, as evidenced by the results in Figures 5, 10, and 11. Let us know if you have any more concerns.
>
> We appreciate the time and effort that Reviewer ebPC has put into evaluating our paper. Your constructive feedback has helped us clarify key aspects of our manuscript.

---

> > ### Comment · Reviewer_ebPC · 2024-08-11
> >
> > I appreciate the detailed response and hard work from the authors.
> >
> > Based on the response and clarifications provided by the authors, I will be raising my score from 4 to 6.
> > My judgement for the score being --- I think the work is fascinating and experiments are good but the connection to RL is rather brief and the performance of the introduced model is decent but not competitive. But I wish the authors best of luck.

---

### Official Review · Reviewer_9C8q · 2024-07-11

**Soundness:** 4
**Presentation:** 3
**Contribution:** 3
**Rating:** 6
**Confidence:** 2

**Summary:**

This paper establishes a direct connection between two important frameworks: neural episodic control and Universal Hopfield Network. It further derives Lyapunov functions for the dynamics and explores the ability of Neural Episodic Control to function as a memory system.

**Strengths:**

The idea of the paper is original and the manuscript is clear. The new theoretical connection is significant and important in relating previously unrelated frameworks.

**Weaknesses:**

While the contribution is relatively important, it is solely based on a mathematical equivalence between the two frameworks. It is a bit unclear to me, whether such an equivalence runs deep and might affect the way memory and RL systems co-function together. In a way, it feels like this paper could dig deeper into an underlying unifying framework.

**Questions:**

- Can the authors further clarify how does the proposed connection between Hopfield Networks and reinforcement learning algorithms advance the current understanding of episodic memory in neural networks?
- In deriving the energy functions, specific modifications were made to the separation function κ. What are the theoretical justifications for these modifications, and how do they impact the overall model performance?
- The paper suggests that the Manhattan distance kernel improves performance over the Euclidean distance kernel. Can you provide a detailed analysis of why this might be the case?
- The introduction of a new criterion to disentangle memorization from generalization is an interesting contribution. How does this criterion compare to existing methods of evaluating associative memory models?

---

> ### Author Rebuttal · Authors · 2024-07-31
>
> We thank Reviewer 9C8q for the constructive feedback and positive remarks on our work. We appreciate the recognition of the originality and clarity of our manuscript, as well as the significance of the theoretical connections made. Below, we address the specific questions and points raised.
>
> 1. Comment: "While the contribution is relatively important, it is solely based on a mathematical equivalence between the two frameworks. It is a bit unclear to me, whether such an equivalence runs deep and might affect the way memory and RL systems co-function together. In a way, it feels like this paper could dig deeper into an underlying unifying framework."
>     - Answer: Thank you for highlighting the importance of our contribution. To better highlight the connection between DND and UHN, we have added a new figure illustrating this equivalence. Our empirical results also provide clear insights into how RL methods based on episodic control could be improved by leveraging this connection. We are currently testing these hypotheses in RL settings, which we believe will further validate and deepen our understanding of this unifying framework.
>
> 2. Question: "Can the authors further clarify how does the proposed connection between Hopfield Networks and reinforcement learning algorithms advance the current understanding of episodic memory in neural networks?"
>     - Answer: The proposed connection allows us to leverage the well-studied dynamics of Hopfield Networks to improve the efficiency and effectiveness of neural episodic control in RL. By understanding how memories are stored and retrieved within this framework, we can design RL systems that better integrate episodic memory, leading to improved decision-making and faster learning. This connection enhances our understanding of episodic memory by providing a unified theoretical framework that explains memory dynamics in both associative memory models and RL systems.
>
> 3. Question: "In deriving the energy functions, specific modifications were made to the separation function [k-max]. What are the theoretical justifications for these modifications, and how do they impact the overall model performance?"
>     - Answer: The modifications to k-max were made to derive the energy functions for neural episodic control. We ensured the new function has nonzero gradient while preserving the functional properties of k-max. As beta and beta_k grow to infinity, the new function approaches k-max. Hence, given sufficently large beta and beta_k, the overall model performance is not affected. We have included this justification in the revised manuscript.
>
> 4. Question: "The paper suggests that the Manhattan distance kernel improves performance over the Euclidean distance kernel. Can you provide a detailed analysis of why this might be the case?"
>     - Answer: Similar to the findings of Millidge et al. (2022) in their Universal Hopfield Network paper, we observe that the Manhattan distance kernel outperforms the Euclidean distance kernel. The selection of similarity functions is crucial because it significantly impacts the ranking of different memories. While the exact reason why the Manhattan distance yields better results is not entirely clear, one possible explanation is that the Euclidean distance involves squaring the differences, which can introduce distortion, whereas the Manhattan distance preserves linearity.
>
> 5. Question: "The introduction of a new criterion to disentangle memorization from generalization is an interesting contribution. How does this criterion compare to existing methods of evaluating associative memory models?"
>     - Answer: We thank the reviewer for highlighting out contribution. Our new criterion provides a more nuanced evaluation by explicitly distinguishing between an associative memory model's ability to memorize specific instances and its ability to generalize across similar instances. We show that the k-max separation function (k>1) can yield state of the art performance with the standard generalization performance. This contradicts the prediction of Millidge et al. (2022) that 1-max is the optimal separation function. We also point out that 1-max is optimal when performance is assessed with the new memorization criterion. Hence, our work challenges the canonical criterion and allows for a richer evaluation of associative memory models.
>
> We hope these clarifications address your concerns and enhance the understanding and impact of our work. Thank you again for your valuable feedback, which has helped improve our manuscript significantly.

---

### Official Review · Reviewer_UfDJ · 2024-07-12

**Soundness:** 3
**Presentation:** 3
**Contribution:** 3
**Rating:** 6
**Confidence:** 1

**Summary:**

The paper introduces the differentiable neural dictionary, which uses template-based memory storage, relating it mathematically to Hopfield Networks within the UHN framework. This novel model is shown to be capable of storing memories through operations such as similarity, separation, and projection, thus demonstrating high capacity and adaptability. The model employs different separation functions like k-nearest neighbor and Max function, with the latter transitioning sharply between memory states upon noise increments. It also discusses how the dictionary outperforms traditional models in associative memory tasks by using different distance kernels (Euclidean, Manhattan), which aids in better generalization over memorization. The discussion extends to the potential applications of DND in episodic control within reinforcement learning, suggesting that DND can speed up learning by reducing the bottleneck in decision processes and integrating generalization into episodic memory. The text ends by suggesting further research into the biological basis of DND and its relationship with neural mechanisms of memory.

**Strengths:**

1. The article is clearly articulated and provides detailed experimental procedures.
2. This paper presents a novel approach to integrating associative memory with reinforcement learning. Specifically, it re-derives mathematical formulations, transforming the form of the Differentiable Neural Dictionary (DND) into the Universal Hopfield Network framework (UHN). This transformation leads to the derivation of corresponding energy functions from the UHN.
3. The paper demonstrates the capacity of the DND model on MNIST, CIFAR-10, and Tiny ImageNet datasets using various functions. It evaluates the model’s retrieval capability against noise, performance based on the memorization criterion, and the relationship between k-Max and Softmax functions.

**Weaknesses:**

The article starts from the Hopfield Network framework, and my concern lies in the fairness of introducing the k-nearest neighbor for comparisons in associative memory. This introduction seems to bring in additional memory information, potentially skewing the comparisons. Furthermore, the paper suggests that evaluating retrieval should involve comparisons with the entire dataset. This approach deviates from the essential nature of memory retrieval tasks. If there are enough elements in the dataset for comparison, such as with the 50-Max criterion, selecting the most likely candidate from a set, even in the presence of significant noise, seems justifiable. However, this method may not accurately reflect the true performance of memory retrieval under more typical conditions where fewer comparison points are available.

**Questions:**

Please see the "Weaknesses" section.

**Limitations:**

The article extensively discusses its limitations towards the end.

---

> ### Author Rebuttal · Authors · 2024-07-31
>
> We thank Reviewer UfDJ for the constructive feedback and positive remarks on our work. We appreciate the recognition of our novel approach and detailed experiments. Below, we address the specific concerns raised.
>
> 1. Comment: "The article starts from the Hopfield Network framework, and my concern lies in the fairness of introducing the k-nearest neighbor for comparisons in associative memory. This introduction seems to bring in additional memory information, potentially skewing the comparisons."
>     - Answer: It is not clear to us what additional memory information is added by introducing the k-nearest neighbor function. If the concern is about memory complexity, it is not impacted by this new separation function. We added information about the computational complexity of the different UHN instances and highlight that the search complexity of k-max is better than other separation functions when implemented with a k-d tree. If the concern is about additional hyperparameters, k is equivalent to softmax's beta, so k-max uses the same number of hyperparameters as softmax.
>
> 2. Comment: "The paper suggests that evaluating retrieval should involve comparisons with the entire dataset. This approach deviates from the essential nature of memory retrieval tasks."
>     - Answer: We do not mean that retrieval should always involve comparisons with the entire dataset. Our intent is to highlight that the standard criterion, which evaluates performance in absolute terms, differs from a criterion that assesses how close the retrieval is to the correct image relative to other images in the training set. This distinction is important as it influences whether the evaluation favors memorization or generalization strategies. We rephrased key sentences in the Discussion section.
>
> We hope these clarifications address your concerns and enhance the understanding and impact of our work. Thank you again for your valuable feedback.

---

> > ### Comment · Reviewer_UfDJ · 2024-08-09
> >
> > Thank you for your response. I have carefully reviewed the other reviewers' comments as well as your replies. I believe I will maintain my current evaluation score.

---

### Official Review · Reviewer_4Gfy · 2024-07-12

**Soundness:** 3
**Presentation:** 3
**Contribution:** 3
**Rating:** 7
**Confidence:** 3

**Summary:**

This paper establishes a novel connection between differentiable neural dictionaries (DNDs) used in episodic control for reinforcement learning and Hopfield networks used as associative memory models. The authors show that DNDs can be formulated within the Universal Hopfield Network (UHN) framework, derive energy functions for DND recall, and conduct experiments comparing DND performance to other associative memory models on image reconstruction tasks.

I believe that this paper presents a valuable theoretical contribution by connecting DNDs to the UHN framework, supported by extensive empirical evaluation. The work has the potential to impact both reinforcement learning and associative memory research. However, the paper would benefit from a bit clearer organization, a more explicit discussion of practical implications, and ideally some exploration of the impact on RL tasks. Finally, the addition of a conceptual figure or schematic would further strengthen the paper by making its key ideas more accessible and memorable. With these improvements, I believe this paper would be a strong contribution to NeurIPS.

**Strengths:**

- Despite some organisation issues I mention below, I still find this paper relatively easy to follow and with good quality of English, plots, and articulation of ideas.
- The theoretical contribution linking DNDs to Hopfield networks and the UHN framework is significant and well-argued. This connection opens up interesting avenues for cross-pollination between reinforcement learning and associative memory research.
- The derivation of energy functions for DND recall, including a novel continuous approximation, is mathematically sound and adds to our theoretical understanding of these models.
- The experimental comparisons are cover three datasets (MNIST, CIFAR10, Tiny ImageNet) and evaluating different similarity and separation functions.
- The introduction of a new "memorization criterion" for evaluating associative memory performance is thoughtful and helps distinguish between memorization and generalization capabilities.

**Weaknesses:**

- While the experiments are comprehensive, they focus solely on image reconstruction tasks. Given the paper's motivation from reinforcement learning, it would have been valuable to include experiments or discussion on how these findings might impact episodic control in RL settings.
- The paper is quite dense and could benefit from clearer organization. Some important findings, like the superior performance of the Manhattan distance similarity function, are buried in the results sections and could be highlighted more prominently.
- The discussion, while interesting, feels somewhat speculative and disconnected from the main technical contributions. This section could be tightened or more clearly linked to the paper's core findings.
- The paper lacks a clear discussion of computational complexity trade-offs between different similarity and separation functions. This would be particularly relevant for practical applications in RL or large-scale associative memory tasks.

**Questions:**

### I have the following questions/suggestions:
- How do you expect the choice of similarity and separation functions to impact sample efficiency and performance in RL tasks using episodic control?
- Have you considered how the memorization vs. generalization trade-off might be dynamically adjusted in a learning system? Could this provide benefits in certain RL scenarios?
- How does the computational cost of DND retrieval with k-Max separation compare to other UHN instances, particularly for large memory sizes?
- Have you considered the relation between this model and other episodic memory-like approaches, including for instance generative models for video, LLMs, and model-based RL?

### Suggestion for improvement:
I think the paper would greatly benefit from the addition of a high-level schematic or conceptual figure that visually illustrates the connection between DNDs and the UHN framework. Such a figure could perhaps
1. Show the parallel structures and operations in DNDs and UHNs side by side.
2. Illustrate how the similarity, separation, and projection operations map between the two frameworks.
3. Visualize the proposed continuous approximation of the k-Max function.
4. Demonstrate how the energy functions relate to the model's dynamics.
A clear, well-designed figure with any (or all) of these, would significantly enhance the paper's accessibility, especially for readers less familiar with either DNDs or UHNs. It would also help to crystallize the paper's main theoretical contribution and make the work more memorable and impactful.

### Minor points:
- The discussion of biological plausibility in relation to hippocampal models adds an interesting interdisciplinary perspective, but it’s very brief. I would extend it if possible.
- The paragraph in line 180 is quite long and difficult to follow. I know you were trying to keep the 9 page limit but in the camera ready version I would recommend you to break it into two paragraphs and rephrase a few sentences.

**Limitations:**

### Overview:
The authors have made an effort to address some limitations of their work, which is commendable. However, there is room for improvement in this area:
1. Scope of experiments: The authors acknowledge that they did not explore the evaluation of similarity and separation functions in reinforcement learning tasks due to time constraints. This is a good start, but they could expand on why this limitation is important and how it might impact the broader applicability of their findings.
2. Theoretical vs. practical implications: While the paper provides a strong theoretical foundation, it could benefit from a more explicit discussion of the limitations in translating these theoretical insights into practical applications, especially in reinforcement learning contexts.
3. Scalability: The paper doesn't adequately address potential limitations in scaling the proposed methods to very large datasets or complex, high-dimensional state spaces that might be encountered in real-world reinforcement learning tasks.
4. Computational resources: There's no discussion of the computational requirements for the various methods compared, which could be a significant limitation in certain applications.
5. Negative societal impact: The authors do not explicitly discuss potential negative societal impacts of their work. While this research is primarily theoretical, it would be beneficial to consider and address potential misuse or unintended consequences of more efficient episodic control in AI systems, in the camera ready version of this paper.
### Suggestions:
1. Include a dedicated "Limitations and Future Work" section.
2. Discuss how to address these limitations in future research.
3. Consider potential societal impacts (e.g., privacy implications, risks of enhanced AI memory systems).
4. Briefly address computational resource requirements and scalability.
Addressing these points would demonstrate a more comprehensive understanding of the work's broader implications and potential impacts.

---

> ### Author Rebuttal · Authors · 2024-07-31
>
> We would like to thank Reviewer 4Gfy for the detailed and constructive feedback on our submission. We appreciate the recognition of our significant theoretical contribution and extensive empirical evaluation. Below, we address each of your comments and suggestions in detail.
>
> 1. Comment: "While the experiments are comprehensive, they focus solely on image reconstruction tasks. Given the paper's motivation from reinforcement learning, it would have been valuable to include experiments or discussion on how these findings might impact episodic control in RL settings."
>     - Answer: We understand your concern. We used three different image datasets, including CIFAR10 and Tiny ImageNet, which are naturalistic and high-dimensional, to evaluate the performance of memory models comprehensively. Image reconstruction tasks provide a robust testbed for assessing memory models due to their complexity and interpretability. We are now working on evaluating the models on RL tasks, which will hopefully give rise to a new manuscript. We have added a paragraph in the Discussion section to outline these future directions.
>
> 2. Comment: "The paper is quite dense and could benefit from clearer organization. Some important findings, like the superior performance of the Manhattan distance similarity function, are buried in the results sections and could be highlighted more prominently."
>     - Answer: We have reorganized the results section to highlight key findings, including the superior performance of the Manhattan distance similarity function, and improved the overall clarity of the manuscript.
>
> 3. Comment: "The discussion, while interesting, feels somewhat speculative and disconnected from the main technical contributions. This section could be tightened or more clearly linked to the paper's core findings."
>     - Answer: We have revised the discussion section to more clearly link it to our core technical contributions, replacing speculative content related to neuroscience by referenced claims and focusing on practical implications and future research directions.
>
> 4. Comment: "The paper lacks a clear discussion of computational complexity trade-offs between different similarity and separation functions. This would be particularly relevant for practical applications in RL or large-scale associative memory tasks."
>     - Answer: We thank the reviewer for raising this very interesting point. We have now provided a comparative analysis of the computational costs for DND retrieval with k-Max separation versus other UHN instances. Neural episodic control implements k-Max using a k-d tree, whose average search complexity is O(log n). In comparison, the complexity of the softmax function as used in modern Hopfield networks is O(n). This difference in complexity can have significant implications for large memory sizes, where the k-Max approach can disregard significant portions of the memory.
>
> 5. Question: "How do you expect the choice of similarity and separation functions to impact sample efficiency and performance in RL tasks using episodic control?"
>     - Answer: We are very much looking forward to finding out how the choice of similarity and separation functions impacts sample efficiency and performance in RL tasks. We believe that functions promoting better memory, such as the Manhattan distance, can enhance sample efficiency by enabling more effective recall of relevant past experiences. We have elaborated on this in the Discussion section and are currently working on experiments to validate these hypotheses in RL settings.
>
> 6. Question: "Have you considered how the memorization vs. generalization trade-off might be dynamically adjusted in a learning system? Could this provide benefits in certain RL scenarios?"
>     - Answer: Yes, gradually shifting from a strategy that memorizes specific examples to one that generalizes aligns with the complementary roles of episodic and semantic memory in cognitive science. We discuss how parameters such as k and beta can be dynamically adjusted during training to balance memorization and generalization. This approach can provide significant benefits by adapting the model to different phases of learning and varying task requirements in reinforcement learning scenarios.
>
> 7. Question: "How does the computational cost of DND retrieval with k-Max separation compare to other UHN instances, particularly for large memory sizes?"
>     - Answer: See 4.
>
> 8. Question: "Have you considered the relation between this model and other episodic memory-like approaches, including generative models for video, LLMs, and model-based RL?"
>     - Answer: We have expanded the related work section to discuss how our model relates to other episodic memory-like approaches, such as generative models for video, large language models (LLMs), and model-based RL, highlighting unique advantages and potential synergies.
>
> 9. Suggestion: "The paper would greatly benefit from the addition of a high-level schematic or conceptual figure."
>     - Answer: We agree and have added a high-level figure that visually illustrates the connection between DNDs and the UHN framework, including parallel structures, operations, and mapping of similarity, separation, and projection functions.
>
> 10. Comment: "The discussion of biological plausibility in relation to hippocampal models is very brief."
>     - Answer: See 3.
>
> 11. Comment: "The paragraph in line 180 is long and difficult to follow."
>     - Answer: We have revised the paragraph, breaking it into two shorter paragraphs and rephrasing sentences for better readability.
>
> 12. Suggestions: "Include a dedicated "Limitations and Future Work" section." and "Consider potential societal impacts"
>     - Answer: We included a "Limitations and Future Work" section and briefly consider societal impacts.
>
> We hope these revisions address your concerns and enhance the quality and impact of our paper. Thank you again for your valuable feedback.

---

> > ### Comment · Reviewer_4Gfy · 2024-08-12
> >
> > Thank you for your detailed and informative response. The revisions you suggest address most of my concerns. I'd prefer if the paper was more complete in terms of including comprehensive RL experiments but I do understand that this topic is so interesting that it might deserve its own paper. After carefully reviewing your responses and the other reviewers' comments, I decided to increase my score to 7. Good luck!

---

### Author Rebuttal · Authors · 2024-07-31

We thank all reviewers for their detailed and constructive feedback on our submission. We are pleased that the reviewers recognize our significant theoretical contribution of linking Differentiable Neural Dictionaries (DNDs) to the Universal Hopfield Network (UHN) framework (R.4Gfy) and the extensive empirical evaluation we conducted (R.4Gfy). The clarity and articulation of our manuscript were also noted positively (R.UfDJ, R.4Gfy, R.9C8q, R.w6Cb, R.ebPC).

Additionally, our novel approach to integrating associative memory with reinforcement learning was appreciated (R.UfDJ), along with the introduction of a new "memorization criterion" (R.4Gfy) and the originality and significance of relating previously unrelated frameworks (R.9C8q). The value of deriving new energy functions for DNDs was also recognized, both theoretically (R.4Gfy) and empirically (R.ebPC).

In response to the reviewers' comments, we have enhanced the clarity, organization, and practical relevance of our manuscript. We addressed concerns about the focus on image reconstruction tasks, provided a clearer discussion of practical implications and computational complexity, and added visual aids to improve accessibility.

In individual rebuttals, we detail our responses to each of the reviewers' comments and outline the revisions made to address their concerns.

---

### Decision · Program_Chairs · 2024-09-25

**Decision:**

Accept (poster)

**Comment:**

The paper connects differentiable dictionaries, used in neural episodic control, with a new model from the Dense Associative Memory family. The paper received favorable reviews and for this reason is accepted. I request that in addition to addressing the issues raised by the reviewers the authors better position their contribution relative to the line of work on Dense Associative Memories (Modern Hopfield Networks).

For instance, the sentence “Some important limitations have been addressed with the development of differentiable continuous Hopfield Networks (2)”  in lines 18, 19 is incorrect. Continuous differentiable Hopfield Networks have been developed by J.Hopfield in 1984 in this paper: https://www.pnas.org/doi/abs/10.1073/pnas.81.10.3088, long before reference (2). The contribution of Ref (2) was to connect the continuous formulation of Modern Hopfield Networks from this paper: https://papers.nips.cc/paper_files/paper/2016/hash/eaae339c4d89fc102edd9dbdb6a28915-Abstract.html with the attention mechanism; and introduce a specific model (from the continuous Modern Hopfield Network family) with a softmax activation. When I read the introduction I am left with an impression that the authors have a limited understanding of the origins of the ideas and methods that they are using in their work. This issue should be addressed in the camera ready version.